# Novel Insight into the Potential Role of Acylglycerophosphate Acyltransferases Family Members on Triacylglycerols Synthesis in Buffalo

**DOI:** 10.3390/ijms23126561

**Published:** 2022-06-12

**Authors:** Xiao-ya Ma, An-qin Duan, Xing-rong Lu, Sha-sha Liang, Pei-hao Sun, Md Mahmodul Hasan Sohel, Hamdy Abdel-Shafy, Ahmed Amin, Ai-xin Liang, Ting-xian Deng

**Affiliations:** 1Key Laboratory of Buffalo Genetics, Breeding and Reproduction Technology, Buffalo Research Institute, Chinese Academy of Agricultural Sciences, Nanning 530001, China; maxiaoya8899@163.com (X.-y.M.); duanaq321@163.com (A.-q.D.); luxingrong074@163.com (X.-r.L.); cangshucangshu@126.com (S.-s.L.); 2Key Laboratory of Agricultural Animal Genetics, Breeding and Reproduction of Ministry of Education, College of Animal Science and Technology, Huazhong Agricultural University, Wuhan 430070, China; sph99@webmail.hzau.edu.cn; 3Department of Genetics, Faculty of Veterinary Medicine, Erciyes University, Kayseri 38039, Turkey; sohel@erciyes.edu.tr; 4Genome and Stem Cell Centre, Erciyes University, Kayseri 38039, Turkey; 5Department of Animal Production, Faculty of Agriculture, Cairo University, El-Gamma Street, Giza 12613, Egypt; hamdyabdelshafy@agr.cu.edu.eg (H.A.-S.); a.amin@agr.cu.edu.eg (A.A.)

**Keywords:** *AGPAT1*, *AGPAT6*, buffalo, functional characterization, mammary epithelial cells

## Abstract

Acylglycerophosphate acyltransferases (AGPATs) are the rate-limiting enzymes for the de novo pathway of triacylglycerols (TAG) synthesis. Although AGPATs have been extensively explored by evolution, expression and functional studies, little is known on functional characterization of how many members of the AGPAT family are involved in TAG synthesis and their impact on the cell proliferation and apoptosis. Here, 13 AGPAT genes in buffalo were identified, of which 12 AGPAT gene pairs were orthologous between buffalo and cattle. Comparative transcriptomic analysis and real-time quantitative reverse transcription PCR (qRT-PCR) further showed that both *AGPAT1* and *AGPAT6* were highly expressed in milk samples of buffalo and cattle during lactation. Knockdown of *AGPAT1* or *AGPAT6* significantly decreased the TAG content of buffalo mammary epithelial cells (BuMECs) and bovine mammary epithelial cells (BoMECs) by regulating lipogenic gene expression (*p* < 0.05). Knockdown of *AGPAT1* or *AGPAT6* inhibited proliferation and apoptosis of BuMECs through the expression of marker genes associated with the proliferation and apoptosis (*p* < 0.05). Our data confirmed that both *AGPAT1* and *AGPAT6* could regulate TAG synthesis and growth of mammary epithelial cells in buffalo. These findings will have important implications for understanding the role of the AGPAT gene in buffalo milk performance.

## 1. Introduction

Water buffalo (*Bubalus bubalis*) is important dairy livestock producing more than 15% of the world’s total milk production. The domestic water buffalo in Asia is generally classified into two major subspecies: River buffalo (2n = 50) is reared mainly for milk, and the swamp buffalo (2n = 48) is primarily raised for draught power [1]. Buffalo milk is a highly nutritious food containing 89% more fat, 28% more protein, and 100% more iron as well as 1% less lactose and 77% less cholesterol content compared to those in dairy cattle [2]. Milk fat is composed of 98% triacylglycerols (TAG), which are the most abundant lipids in humans and animals. It is well reported that TAG determines the functional and physical properties of several dairy products such as the spreadability of butter [3]. TAG is formed by a combination between glycerol and three fatty acids. The availability of fatty acids present in mammary epithelial cells along with enzymes involved in the metabolism largely affects the characteristics of TAG, which is partially controlled by several genetic factors [4,5]. The synthesis of TAG in eukaryotes is controlled by two major pathways: the Kennedy pathway (*de novo* pathway) and the monoacylglycerol pathway. The *de novo* pathway is the major pathway for TAG synthesis performed in most mammalian cells. Evidence showed that four rate-limiting steps play a vital role in determining the triglyceride content [6], which are controlled by the glycerophosphate acyltransferase (GPAT), acylglycerophosphate acyltransferase (AGPAT), lipid phosphate phosphohydrolase (LPIN), and diacylglycerol acyltransferase (DGAT) gene families, respectively [7].

The AGPAT family acts on the second acylation step in the de novo pathway to catalyze the conversion of lysophosphatidic acid (LPA) to phosphatidic acid (PA). To date, genome-wide identification, evolution, gene structure, and expression profile analyses have identified eleven gene members (AGPAT1-11) of the AGPAT gene family in different species [8,9,10]. Evidence revealed that the AGPAT family was ancient and experienced different origins, with many eukaryotic species having multiple genes originated by duplication events [9]. For example, the duplication events resulted in the origination of two isoforms *AGPAT3* and *AGPAT4* in animal species. Gene structure revealed that the number of introns per gene varied from one to thirteen [11]. In addition, four acyltransferase motifs (motifs I~IV) were also found in the animal species. Among them, motifs I and IV contain a conserved NHxxxxD and proline sequence, respectively, involving in the acyl-CoA binding and catalysis. Motifs II and III are involved in LPA binding, with a conserved arginine and Glutamate–Glycine–Threonine–Arginine (EGTR) sequence, respectively [7,8,11]. Moreover, numerous studies have characterized the expression profiles of the AGPAT gene family in different animals based on different datasets, such as the microarray expression data of humans and mice [9], RNA-seq data of chicken [10], and real-time quantitative PCR (qRT-PCR) data for cattle and pigs [12,13], implying that different members had spatial and temporal specific expression patterns in different tissues. For example, *AGPAT6* (Known as *GPAT4*) appears to be the most abundant AGPAT isoform in the bovine mammary gland, followed by *AGPAT1* [12]. Numerous studies also found that *AGPAT6* polymorphisms were highly associated with milk production traits in cattle including milk fat percentage [14], fat compositions [15], and fatty acids [16]. Notably, Nafikov et al. [17] also found that the haplotypes of the *AGPAT1* gene were associated with higher polyunsaturated fatty acids (PUFA) and linoleic acid concentrations in cattle. These findings suggested that both *AGPAT1* and *AGPAT6* play a vital role in TAG synthesis. However, information on how many members of the AGPAT family are involved in the TAG synthesis of mammary epithelial cells is still extremely limited. Since the milk production of dairy cows is significantly higher than that of buffaloes and the functional importance of AGPATs, it is of considerable importance to characterize the function of buffalo AGPAT members on TAG synthesis. In addition, the identification of excellent functional genes will be of great significance to the molecular breeding practices of this species. In this regard, taking dairy cattle as a reference, the objectives of this study were first to characterize the differential expression pattern of the AGPAT gene family in both cattle and buffalo using comparative transcriptome analysis; subsequently, to investigate the potential functional role of AGPAT genes on cell growth and TAG synthesis in buffalo mammary epithelial cells (BuMEC) and bovine mammary epithelial cells (BoMEC) using RNA interference (RNAi) technology.

## 2. Results

### 2.1. Identification and Sequence Analysis of Buffalo AGPAT Genes

In total, 32 and 14 AGPAT isoform protein sequences encoded by 13 AGPAT genes were predicted from the river and swamp buffalo genome, respectively (Appendix A). The open reading frames (ORFs) of buffalo AGPAT protein isoforms ranged from 762 to 2136 bp in length when encoding the protein of 253 to 711 residues, with a predicted MW from 28.93 to 78.25 kDa. The pI values of these isoforms ranged from 6.20 to 11.26. Phylogenetic analysis revealed that all buffalo AGPAT genes could be divided into four clusters (Cluster I, II, III, and IV) containing 4, 2, 2, and 5 genes, respectively (Figure 1). Cluster IV was the larger one with the 5 members of AGPATs, followed by Cluster I with 4 AGPAT genes, while Cluster II and III were the smaller ones (*n* = 2). The constructed dendrogram further showed that the buffalo AGPAT gene family was usually the most closely evolutionary relationship with the other five representative mammals (Cattle, Goat, Sheep, Horse, and Human). Moreover, the motif analysis showed that a total of 10 conserved motifs were detected in the identified buffalo AGPAT genes (Appendix A). Here, four motifs (MEME-1, MEME-3, MEME-5, and MEME-7) were annotated as the collagen domain after the Pfam search (Appendix A). Interestingly, we observed that the AGPATs in Cluster I had three acyltransferase domains (MEME-1, MEME-3, and MEME-7) and one EF-hand_1 domain. The AGPATs in remaining Cluster II and III had two acyltransferase domains with the MEME-1 and MEME-3 motifs, but Cluster II AGPATs had an EF-hand_1 domain. Cluster IV AGPATs had MEME-3 and MEME-5 acyltransferase domains. Moreover, conserved protein domain analysis revealed that a total of eight domains were found in the analyzed AGPAT protein sequences (Appendix A). AGPATs in the Cluster I had an LPLAT_LPCAT1-like conserved domain. AGPATs in Cluster II and III had acyltransferase C-terminus and LPLAT_AGPAT-like domains, respectively. AGPATs in Cluster IV had LPLAT_LPCAT1-like and AGPAT conserved domains.

### 2.2. Comparative Transcriptomic Analyses of Orthologous AGPATs between Buffalo and Cattle

Prior to the comparative transcriptomic analysis of the AGPAT gene family between buffalo and cattle, we first performed a collinearity analysis between the two species. The chromosomal mapping revealed that a total of 13 and 12 AGPAT genes were found to be randomly distributed on 10 chromosomes, which are mainly located on the proximate or the distal ends of the chromosomes in the buffalo and cattle, respectively (Appendix A). Collinearity analysis showed that 12 AGPAT gene pairs were orthologous between the two species (Figure 2A). The divergence times of all orthologous gene pairs between buffalo and cattle ranged from 0.465 to 2.937 Mya. All orthologous gene pairs had Ka/Ks ratios that were less than 0.5 (Appendix A).

Using the RNA-seq data from the milk samples, we observed that three AGPAT genes (*AGPAT1*, *AGPAT6*, and *LPCAT1*) were highly expressed in the three lactation stages (early-, mid-, and late-lactation) in buffalo milk (Figure 2C). By contrast, two AGPAT genes (*AGPAT1* and *AGPAT6*) were found to be highly expressed in the same three lactation stages in cattle milk (Figure 2B). Interestingly, the expression pattern of *AGPAT1* and *AGPAT6* genes existed in a complementation relationship in buffalo during lactation. In cattle, the *AGPAT6* gene was always highly expressed during lactation, followed by the *AGPAT1* gene. Moreover, gene expression analysis by using the qRT-PCR test also showed that both *AGPAT1* and *AGPAT6* were highly expressed in mammary gland tissues (Figure 2D). It can be inferred that both *AGPAT1* and *AGPAT6* genes might have an important influence on milk production performance for both buffalo and cattle.

### 2.3. Knockdown of AGPAT1 and AGPAT6 Decreased TAG Concentration in BuMECs and BoMECs

For the further exploration of the potential impact of *AGPAT1* and *AGPAT6* on milk performance in both buffalo and cattle, we first investigated the TAG content of BuMECs and BoMECs after silencing the two genes for 48 h. As shown in Figure 3A, the interference efficiency of *AGPAT1* and *AGPAT6* in the BuMECs was 95% and 76% (*p* < 0.01), while their interference efficiency in BoMECs was 89% and 82% (*p* < 0.01), respectively. The results suggested that these siRNA fragments could be used for further analysis. Subsequently, we observed that knockdown of *AGPAT1* (*p* < 0.01) or *AGPAT6* (*p* < 0.05) genes significantly decreased TGA concentration in BuMECs and BoMECs (Figure 3B,C). We found that *AGPAT1* knockdown in both BuMECs and BoMECs decreased the mRNA expression levels of lipogenic pathway-related genes (Fatty acid synthase (*FASN*), Acyl-CoA Synthetase Long Chain Family Member 1 (*ACSL1*), Glycerol-3-Phosphate Acyltransferase Mitochondrial (*GPAM*), Lipin 1 (*LPIN1*), desaturation-related genes (*SCD*), Diacylglycerol O-Acyltransferase 1(*DGAT1*), and Perilipin 2 (*PLIN2*)) while increasing the expression level of Acetyl-CoA Carboxylase Alpha (*ACACA*) (Figure 3D; *p* < 0.05). As for the knockdown of the AGPAT6 gene, we observed that 3 fatty acid synthesis-related genes (*FASN*, *ACSL1*, and *ACACA*) had higher expression levels compared to the control group, while another 5 TAG synthesis-related genes (*SCD, GPAM*, *LPIN1*, *DGAT1*, and *PLIN2*) had lower mRNA expression than that of the control (Figure 3E; *p* < 0.05).

### 2.4. Knockdown of AGPAT1 and AGPAT6 Suppress BuMECs Proliferation

The effect of *AGPAT1* and *AGPAT6* gene knockdown on BuMECs proliferation was investigated at three time points (24, 48, and 72 h). Results showed that knockdown of *AGPAT1* and *AGPAT6* in BuMECs decreased the cell population (*p* < 0.05) in the culture medium compared to the control group (Figure 4A,C). Moreover, the mRNA levels of two proliferation-related genes (Tumor Protein P53 (*TP53*) and *CyclinD1*) were further determined after *AGPAT1* or *AGPAT6* knockdown using qRT-PCR. The results showed that increases in the mRNA expression level of the *TP53* gene were observed after the *AGPAT1* (Figure 4B; *p* < 0.001) and *AGPAT6* (Figure 4D; *p* < 0.01) silencing. For *CyclinD1*, a significant level was observed after *AGPAT1* (Figure 4B) and *AGPAT6* (Figure 4D) knockdown (*p* < 0.05).

### 2.5. Knockdown of AGPAT1 and AGPAT6 Inhibits BuMECs Apoptosis

Using the Annexin V-EGFP Kit, we observed the apoptosis rate of the BuMECs after *AGPAT1* and/or *AGPAT6* gene knockdown. The results showed that the apoptosis rate decreased in BuMECs after *AGPAT1* or/and *AGPAT6* silencing for 48 h (Figure 5A–C). mRNA expression analysis showed that both *AGPAT1* and *AGPAT6* had lower expression levels than that of the control group (Figure 5D; *p* < 0.05). Moreover, the expression of four marker genes (BCL2 Associated X, Apoptosis Regulator (*BAX*), Fas Cell Surface Death Receptor (*FAS*), BCL2 Apoptosis Regulator (*BCL-2*), and *Caspasse6*) related to apoptosis was determined using qRT-PCR in BuMECs following treatments with si-AGAPT1 and si-AGPAT6. The expression level of *BAX*, *FAS*, and *BCL-2* was 0.765, 0.830, and 0.619 in the si-AGPAT1 group, respectively, decreasing by 23.50%, 17.00%, and 28.10% in the control group; correspondingly, the expression level of *Caspasse6* gene was 1.580, increasing by 58.0% in the control group (Figure 5E). Similar results were observed in the si-AGPAT6 group (Figure 5F) and si-AGPAT1/6 group (Figure 5G).

## 3. Discussion

In the present study, a total of 13 AGPAT genes were identified in the river and swamp buffaloes. The identified AGPAT genes were unevenly distributed on the proximate or the distal ends of 10 chromosomes in buffalo. The members of buffalo AGPAT gene family were divided into four clusters based on their phylogenetic relationships. The phylogenetic classification of buffalo AGPAT gene family was also supported by conserved motif and gene structure analyses. Four highly conserved motifs were observed in the AGPAT family. The members in cluster I had three acyltransferase motifs and one EF-hand_1 domain. Two acyltransferase motifs were observed in Cluster II, III, and IV. The conserved protein domain analysis also supported this point. In addition, most closely related members in the same cluster of the AGPAT family harbored similar intron–exon structures.

The expression patterns of orthologous genes are often conserved and closely related to their function [18,19,20]. In the current study, we observed a total of 12 AGPAT genes orthologous between buffalo and cattle, suggesting that they might have a similar function in both species. We found that the *LPCAT1* gene has a higher expression level in buffalo milk than that in dairy cows, which may explain the difference in milk fat content between the two species. However, this needs further experimental verification. Moreover, our data showed that both *AGPAT6* and *AGPAT1* were highly expressed in milk samples during lactation and mammary gland tissue in buffalo. A previous study reported that the *AGPAT6* gene was the most abundant isoform in mammary gland tissue in cattle, accounting for 60% of all AGPAT mRNA, followed by *AGPAT1* (18%) and *AGPAT3* (10%) (Bionaz and Loor, 2008a). Another study revealed that *AGPAT6* was also highly expressed in buffalo mammary gland tissues [21]. These results suggested that the two AGPAT members might play a dominant role in the milk fat synthesis pathway.

To further confirm this hypothesis, we explored the potential function of *AGPAT1* and *AGPAT6* genes on milk fat synthesis in BuMECs and BoMECs. Knockdown of *AGPAT1* or *AGPAT6* genes significantly decreased TAG concentrations in BuMECs and BoMECs (*p* < 0.05). We observed that the TAG content in BuMECs and BoMECs decreased by approximately 60.8% and 61.40% after *AGPAT1* knockdown, respectively (Figure 3B). On the other hand, the knockdown of *AGPAT6* resulted in an approximately 20% and 17% decrease in TAG content of BuMECs and BoMECs, respectively (Figure 3C). The reasons for the discrepancy may be caused by the fact the *AGPAT1* and *AGPAT6* genes have functional differences in regulating milk fat synthesis. Our data revealed that the knockdown of *AGPAT1* and *AGPAT6* in both BuMECs and BoMECs significantly decreased in the mRNA expression levels of the fatty acid synthesis and desaturation-related gene (*SCD*), TAG synthesis-related genes (*GPAM*, *DGAT1*, and *LPIN1*), and lipid droplet formation-related gene (*PLIN2*). These results indicated that both *AGPAT1* and *AGPAT6* genes could decrease TAG content by regulating lipogenic genes. Moreover, we observed that two genes (*FASN* and *ACSL1*) had differential mRNA expression levels in BuMECs after *AGPAT1* or *AGPAT6* knockdown. They had upward trend expression levels after *AGPAT6* knockdown compared to the control group, which was the opposite effect in the knockdown treatment of *AGPAT1*. Evidence showed that both *FASN* and *ACSL1* genes are involved in fatty acid synthesis and play a role in activating fatty acids destined for TAG synthesis [22,23,24]. More importantly, similar results were also found in BoMECs. These results indicated that the *AGPAT1* and *AGPAT6* genes had functional differences in regulating fatty acids synthesis. In other words, *AGPAT1* knockdown inhibited fatty acids synthesis in contrast to the knockdown of *AGPAT6*, which promoted fatty acids synthesis. As it is well known that triacylglycerol composes one glycerol and three fatty acids, the knockdown of *AGPAT6* alleviated the decrease in TAG concentration to some extent. This explains why the reduction rate in TAG content by the treatment of *AGAPT6* knockdown is lower than that of *AGPAT1*. In short, these findings suggested that the role of *AGPAT1* and *AGPAT6* genes in determining milk fat synthesis is not only reflected in the lipogenic genes of the *de novo* pathway but also in their effect on fatty acid synthesis.

An increasing amount of experimental evidence demonstrated that the proliferation and apoptosis of mammary epithelial cells had influences on the development of the mammary gland, milk secretion, and lactation [25,26,27]. Using the siRNA strategy, we found that both *AGPAT1* and *AGPAT6* knockdown could suppress proliferation and apoptosis of BuMECs. This occurs because the knockout of *AGPAT1* and *AGPAT6* may alter the synthesis of LPA, which plays a substantial role in cell proliferation and apoptosis [28,29,30]. Yuh [31] demonstrated that LPA regulates mammary epithelial cell growth and morphology not only by a direct LPA receptor-mediated pathway but also an indirect cellular signal pathway such as the transactivation of EGFR tyrosine kinase. Thus, these results suggested that both *AGPAT1* and *AGPAT6* genes might regulate buffalo mammary epithelial cell growth through the alteration of LPA expression. To further confirm this hypothesis, several genes related to proliferation and apoptosis were selected and used for qRT-PCR analysis. Evidence showed that the stronger expressions of *TP53* and *CyclinD1* were associated with the inhibition [32] and promotion [33] of cell proliferation, respectively. The results suggested that both *AGPAT1* and *AGPAT6* inhibited BuMEC proliferation by upregulating the mRNA expression of *TP53* and downregulating the *CyclinD1* expression. We also observed that both *AGPAT1* and *AGPAT6* reduced cell viability. It is well known that the four genes (*BAX*, *FAS*, *BCL-2*, and *Caspasse6*) are associated with cell apoptosis [28,34,35]. Interestingly, our data revealed that the mRNA expression of the selected marker genes of cell apoptosis also illustrated the same results. All these results provided evidence that the *AGPAT1* or *AGPAT6* gene could suppress the apoptosis of BuMECs by downregulating *BAX*, *FAS*, and *BCL-2* genes and upregulating the *Caspasse6* gene at the mRNA expression level.

## 4. Materials and Methods

### 4.1. Identification and Sequence Analysis of Buffalo AGPAT Genes

To perform the genome-wide identification of AGPAT genes in buffalo, we downloaded public genome datasets of seven representative mammals, including human (*GRCh38.p12*), cattle (*ARS-UCD1.2*), river buffalo (*UOA_WB_1*), goat (*ARS1*), sheep (*Oar_rambouillet_v1.0*) and horse (*EquCab3.0*) datasets from the National Center for Biotechnology Information (NCBI) Genome database, as well as for swamp buffalo (accession number: GWHAAJZ00000000) deposited in the Genome Warehouse in the Beijing Institute of Genomics (BIG) Data Center. The known sequences of some AGPAT proteins were downloaded from the UniPort [36] and used to build the hidden Markov model (HMM) profile with HMMER v3.3.1 software [37]. All AGPAT proteins were examined in the studied species by using HMMER v3.3.1 software [37] with default parameters. Subsequently, these protein sequences were further used for multiple alignments and phylogeny tree construction implemented in MEGA-X [38] software. Identified AGPAT protein sequences were subjected to the ExPASy proteomics server for obtaining molecular weight (MW) and isoelectric points (pI). The motif patterns and element annotations of AGPATs were analyzed using the MEME platform and Pfam programs, respectively. The conserved domain of buffalo AGPATs was predicted using CDD tools of NCBI. The motif pattern and conserved domains of buffalo AGPATs were visualized by TBtools v1.051 software [39].

### 4.2. Chromosomal Mapping and Collinearity Analysis for AGPAT Family

Chromosome locations of the AGPAT gene family in buffalo and cattle were obtained from their genome resources. The identification of orthologous AGPAT genes between buffalo and cattle was analyzed and visualized by TBtools v1.051 software [39] with one-step MCScanX command. Synonymous (Ks) to nonsynonymous (Ka) substitution ratio (Ka/Ks) for orthologous AGPAT pairs was calculated by TBtools v1.051 software [39]. The divergence time for each orthologous gene pair was evaluated by T = Ks/2λ × 10^−6^ million years ago (Mya) [40], where T is the absolute time of divergence, Ks is the synonymous substitution rate, and λ is the clock-like rates in buffalo of 1.26 × 10^−8^ [41].

### 4.3. Comparative Transcriptomic Analyses for Orthologous AGPAT Genes

To explore the differential expression of the AGPAT gene family between buffalo and cattle, two published RNA-seq data (BioProject: PRJNA419906 and PRJNA453843; both from the NCBI SRA database) from milk samples were utilized for conducting comparative transcriptome analyses. Overall, controlling the quality of raw data was performed by the Trim galore ver0.6.6 software. Mapping of the cleaned data from buffalo and cattle was conducted by the HISAT ver.2.2.1 software [42], corresponding to UOA_WB_1 and ARS-UCD1.2 versions as reference genomes, respectively. The count matrix of gene or transcript was constructed by StringTie ver.2.1.4 software [43]. Transcripts per million (TPM) values for each gene were obtained using the DESeq2 R-package [44]. Finally, the differential expression analysis of orthologous AGPATs between buffalo and cattle milk samples was performed. Clustering and generation of a heat map of TPM values for the selected genes were performed using the TBtools v1.051 software [39].

### 4.4. Cell Culture and Transfection

Both BuMECs and BoMECs were preserved in the Buffalo Research Institute laboratory. BuMECs were cultured in DMEM/F-12 medium (Thermo Fisher Scientific, San Diego, CA, USA) supplemented with 10% fetal bovine serum, 5 μg/mL insulin, 1 μg/mL progesterone, 1 μg/mL Hydrocortisone, 5 μg/mL prolactin (Sigma-Aldrich, St. Louis, MO, USA), and 10 ng/mL epidermal growth factor 1 (Thermo Fisher Scientific, San Diego, CA, USA) and incubated at 37 °C in a humidified atmosphere of 5% CO_2_ level. At the same time, BoMECs were also cultured under the same cultural conditions. The siRNA fragments for *AGPAT1* and *AGPAT6* were designed and synthesized by GenePharma (Shanghai, China). These siRNA fragment sequences are listed in Appendix A. Both BuMECs and BoMECs were either transfected with AGPATs siRNAs (siAGPATs) or negative control (NC) siRNAs using RNAiMAX following the manufacturer’s protocol (Thermo Fisher Scientific, San Diego, CA, USA), respectively.

### 4.5. Cell Viability Assay

Cell viability was detected after the knockdown of *AGPAT1* or *AGPAT6* using CellTiter-Glo Reagent (Promega, Madison, WI, USA) according to the manufacturer’s instructions. Briefly, cells were seeded in a 96-well culture plate. At 60% confluency, cells were transfected. After treatments with three time points (24 h, 48 h, and 72 h), cells were then equilibrated at room temperature for 30 min. Subsequently, 100 μL of compound reagents was added to 100 μL of medium-containing cells, mixed for 2 min on an orbital shaker to induce cell lysis, and incubated at room temperature for 10 min to stabilize the luminescent signal. Control wells containing medium without cells were also prepared to obtain a value for background luminescence, and this was followed by luminescence measurement.

### 4.6. Cell Apoptosis Assay

A cell apoptosis assay was performed using Annexin V-EGFP Apoptosis Detection kit (Abcam, Boston, MA, USA). According to the manufacturer’s protocol, the cells were collected by centrifugation, resuspended in 500 μL of binding buffer, added to 5 μL of Annexin V-EGFP and 5 μL of propidium iodide, and incubated at room temperature for 5 min in the dark. After that, the cell suspension was placed on a glass slide, and results were detected under the fluorescence microscope, where the cells that bound Annexin V-EGFP were stained with green in the plasma membrane, while cells that have lost membrane integrity were shown in red staining through the nucleus and a halo of green staining (EGFP) on the cell surface.

### 4.7. Triglyceride Assay

Cellular TAG content in BuMECs and BoMECs was evaluated by using a TAG assay kit (Applygen, Beijing, China). Briefly, cells were seeded in 24 well plates and cultured until 60–70% confluency. Subsequently, the cultured cells were transfected with siRNA fragments. After 48 h, the culture medium was removed and the cells were washed three times with 0.01 mol/L PBS. The TAG level was calibrated with protein concentrations determined with a BCA protein assay kit (Applygen, Beijing, China) and expressed as total TAG per cellar protein. Each experiment was performed in triplicate and repeated at least three times.

### 4.8. Quantitative Real-Time PCR

Total RNA was isolated and purified with PureLinkTM RNA Mini Kit (Thermo Fisher Scientific, San Diego, CA, USA). After that, the First Strand cDNA Synthesis Kit (Thermo Fisher Scientific, San Diego, CA, USA) was used to synthesize the first-strand of cDNA. A LightCycler 480 II sequence detection system instrument (Roche, Vienna, Austria) was used to quantify the transcript abundance of the selected genes. The qRT-PCR reactions were set up in 20 μL containing 10 μL Power-up SYBR Green Master Mix (Thermo Fisher Scientific, San Diego, CA, USA), 2 μL first-strand cDNA template, 0.3 μM of forward and reverse gene-specific primers, and 7.4 μL deionized H_2_O. Expression analysis was performed using a comparative CT(2-∆∆Ct) method [45]. The *GAPDH* gene was used as an endogenous control. All the primers used for qRT-PCR are listed in Appendix A.

### 4.9. Statistical Analysis

Statistical analyses were conducted using GraphPad Prism 9.0 software [46]. The analyzed data was presented as the mean and standard error of the mean (SEM). Each test was performed in triplicate. Statistical significance between the contrasting groups was determined by Student’s *t*-test or one-way analysis of variance (ANOVA). The significance level was declared at *p* < 0.05.

## 5. Conclusions

According to the findings herein obtained, a total of 13 AGPAT genes were found in buffalo, of which 12 orthologous gene pairs were observed between buffalo and cattle. Furthermore, both *AGPAT1* and *AGPAT6* were highly expressed in milk samples in buffalo and cattle during lactation. According to the in vitro assay applied in the study, both *AGPAT1* and *AGPAT6* genes were not only regulating TAG synthesis in mammary epithelial cells but also affected their growth (Figure 6). These findings herein gathered provided new insight into the AGPAT family members on how they regulate TAG synthesis and the growth of mammary epithelial cells in buffalo.

## Figures and Tables

**Figure 1 ijms-23-06561-f001:**
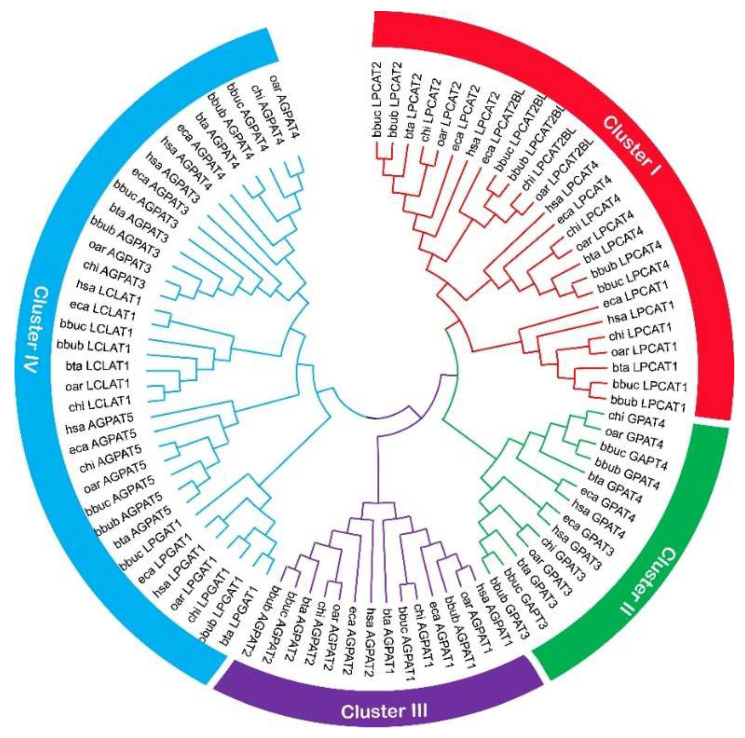
Phylogenetic relationship of AGPAT proteins in seven representative mammals. Line or circle with different colors indicates different clusters. River buffalo: bbub; swamp buffalo: bbuc; cattle: bta; goat: chi; sheep: oar; horse: eca; human: hsa.

**Figure 2 ijms-23-06561-f002:**
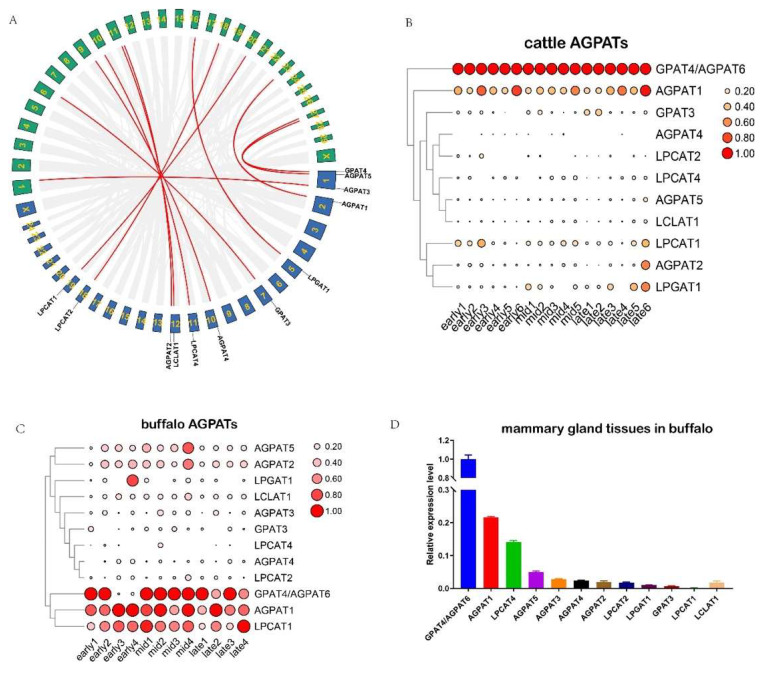
Collinearity and expression analysis of AGPAT family between buffalo and cattle. (**A**) Collinearity analysis of AGPAT family between buffalo and cattle. The green frame indicates cattle chromosomes, the blue frame indicates buffalo chromosomes, and the red line represents collinearity gene pairs. (**B**) Transcriptome analysis of cattle AGPAT family. The area indicates the scale size. (**C**) Transcriptome analysis of buffalo AGPAT family. The area indicates the scale size. (**D**) Expression analysis of buffalo AGPAT genes in mammary gland tissue.

**Figure 3 ijms-23-06561-f003:**
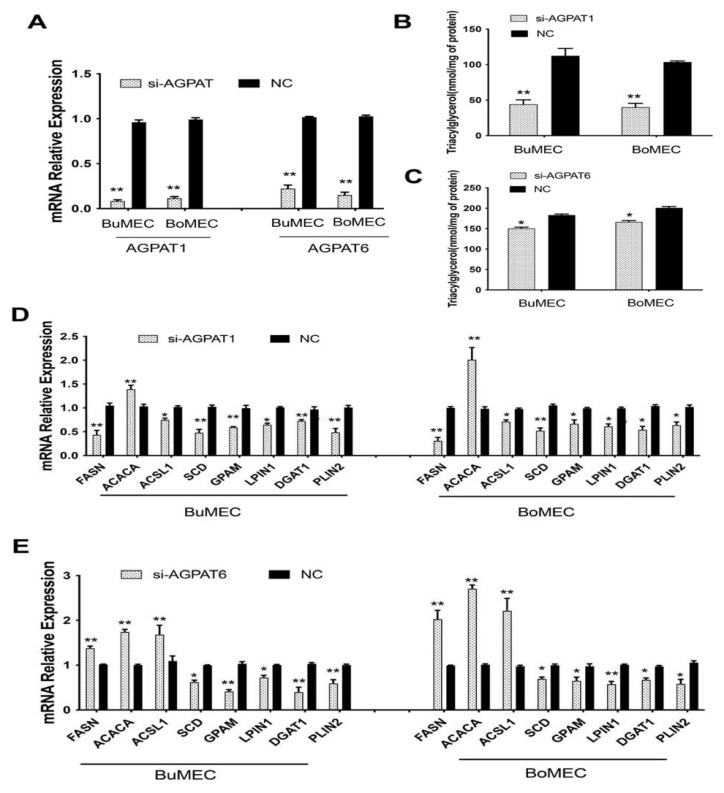
AGPAT1 and AGPAT6 regulated the TAG synthesis of BuMECs and BoMECs. (**A**) The interference efficiency analysis of *AGPAT1* and *AGPAT6* in BuMECs and BoMECs, respectively; (**B**) Effect of *AGPAT1* knockdown on the TAG content of BuMEC and BoMECs; (**C**) Effect of *AGPAT6* knockdown on the TAG content of BuMEC and BoMECs; (**D**) effect of *AGPAT1* knockdown on the expression of genes related to the TAG synthesis in BuMEC and BoMECs, respectively; (**E**) Effect of *AGPAT6* knockdown on the expression of genes related to TAG synthesis in BuMEC and BoMECs, respectively. * *p* < 0.05; ** *p* < 0.01. si-AGPAT includes the siRNA interference for *AGPAT1* or siRNA interference for *AGPAT6*; si-AGPAT1 represents the siRNA interference for *AGPAT1*; si-AGPAT6 represents the siRNA interference for *AGPAT6*; NC indicates the control group.

**Figure 4 ijms-23-06561-f004:**
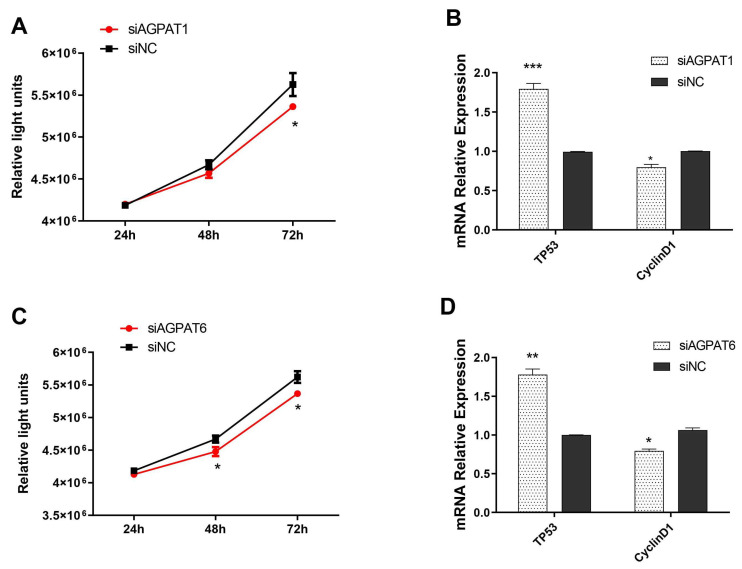
AGPAT1 and AGPAT6 induced cell proliferation of BuMECs. (**A**) Effect of *AGPAT1* knockdown on BuMECs by the CCK-8 detection; (**B**) effect of *AGPAT1* knockdown on proliferation-related gene expression; (**C**) Effect of *AGPAT6* knockdown on BuMECs by the CCK-8 detection; (**D**) Effect of *AGPAT6* knockdown on proliferation-related genes expression. * *p* < 0.05; ** *p* < 0.01; *** *p* < 0.001. si-AGPAT1 represents the siRNA interference for *AGPAT1*; si-AGPAT6 represents the siRNA interference for *AGPAT6*; NC indicates the control group.

**Figure 5 ijms-23-06561-f005:**
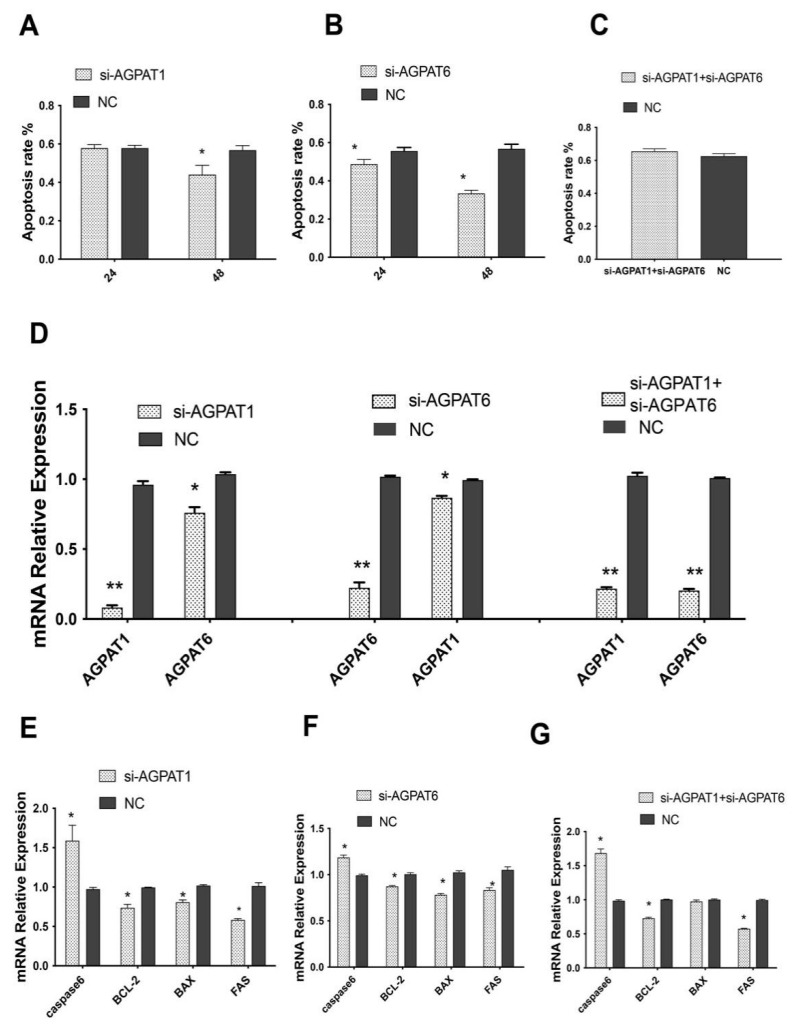
AGPAT1 and AGPAT6 promote cell apoptosis of BuMECs. (**A**) Apoptosis rates detection of *AGPAT1* knockdown in BuMECs; (**B**) Apoptosis rates detection of *AGPAT6* knockdown in BuMECs; (**C**) Apoptosis rates detection of both *AGPAT1* and *AGPAT6* knockdown in BuMECs; (**D**) The expression levels of *AGPAT1* and *AGPAT6* after *AGPAT1* or/and *AGPAT6* silencing at 48 h; (**E**) Effect of *AGPAT1* knockdown on apoptosis-related genes expression; (**F**) effect of *AGPAT6* knockdown on apoptosis-related genes expression; (**G**) Effect of both *AGPA1* and *AGPAT6* knockdown on apoptosis-related genes expression. * *p* < 0.05; ** *p* < 0.01. si-AGPAT1 represents the siRNA interference for *AGPAT1*; si-AGPAT6 represents the siRNA interference for *AGPAT6*; NC indicates the control group.

**Figure 6 ijms-23-06561-f006:**
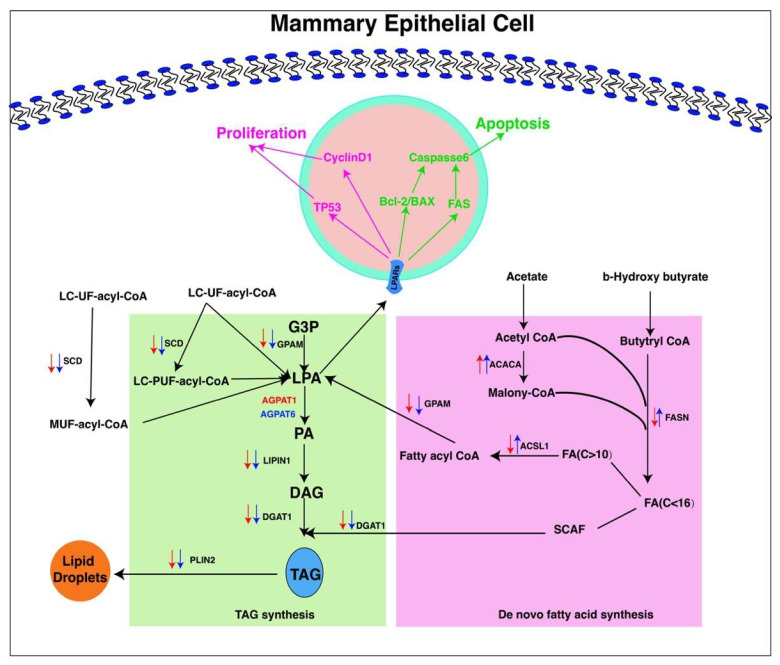
Schematic illustration for regulation mechanism of both *AGPAT1* and *AGPAT6* genes on TAG synthesis, apoptosis, and proliferation. Red arrow represents the expression of lipid metabolism related genes regulated by *A**GPAT**1*; blue arrow represents the expression of lipid metabolism related genes regulated by *A**GPAT6*; purple font and arrows represent cell proliferation involving both *AGPAT1* and *AGPAT6* genes; Green font and arrows represent cell apoptosis involving both *AGPAT1* and *AGPAT6* genes.

## Data Availability

Not applicable.

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
