# Peer review of "Novel Insight into the Potential Role of Acylglycerophosphate Acyltransferases Family Members on Triacylglycerols Synthesis in Buffalo"

_ijms, 2022, doi:10.3390/ijms23126561_

Round 1

Reviewer 1 Report

The title well reflects the major findings of the study. However, I suggest to specify the full term of acronym “AGPAT” as “Acylglycerophosphate acyltransferases”

Please check the punctuation throughout the text.

The abstract adequately summarize methodology, results, and significance of the study. However, Authors should indicate the statistical analysis applied on the obtained data as well as the results together with P values.   

Please rewrote the sentence “Our data provide new insights into the AGPAT family members on how they regulate the TAG synthesis and growth of mammary epithelial cells in buffalo.”

The introduction section is well written and it falls within the topic of the study. However, more information regarding the peculiarity of lactation phase in livestock should be added. Authors should enhance the first sentence by adding more information regarding physiological adaptation of mammals  during life phases as peripartum and reproduction activity.  On this regards, at the beginning of introduction, I suggest to add "During particular life phase such as peripartum females mammals have to face with physiological body demands and adaptation. Particularly, lactation is a very demanding period for the animal, which requires a significant effort of the organism to meet all needs, but also the effort of farmers to provide those needs. Despite the action of homeostatic mechanisms to maintain blood parameters within physiologic levels, changes in metabolites and hormones occur as a result of increased metabolic demands in lactating animals. These changes are not necessarily indicative of diseases but make animals physiologically unstable and more susceptible to a number of metabolic diseases at this stage than during other life periods compromising productivity (Piccione G. et al., Animal Science Paper and Reports, 2009, 27 (4): 321-330; Gianesella M., et al., Journal of Dairy Research, 2019, 86 (3): 291-295; Arfuso F., et al., Theriogenology, 2016, 86: 1156-1164; Fiore E., et al., PLoS One, 2018, 13(4): e0193803; Fiore E., et al. Animal Production Science, 2017, 57 (6): 1007-1013; Morgante M., et al., Animal Science Journal, 2010, 81: 722-730.; Arfuso et al., Archives Animal Breeding, 2016, 59: 429-434)."

The section of Materials and Methods is clear for the reader and it meticulously describes the methods applied in the study.  However, Authors should check this section and correct many punctuation errors.

Regarding statistical analysis, did the Authors apply a normality test on data in order to assess their normal distribution? Please clarify this aspect.

Results section as well as Discussion section is clear and well written. The findings obtained in the study were well discussed and justified with appropriate references.

The conclusion section is well written, Authors well summarize the results and the significance of the study. However, I suggest to avoid making references to figures in this section and the use of personal form (i.e. our, we…).

Please delete “In conclusion…” at the beginning of section.

I suggest to rewrote the sentence as “According to the findings herein obtained, a total of 13 AGPAT genes were found in buffalo, of which 12 orthologous gene pairs were observed between buffalo and cattle. Furthermore, both AGPAT1 and AGPAT6 were highly expressed in milk samples in buffalo and cattle during lactation. According to in vitro assay applied in the study, both AGPAT1 and AGPAT6 genes are not just regulating the TAG synthesis in mammary epithelial cells but also affecting their growth. These findings herein gathered provided new insight into the role of AGPAT family members on TAG synthesis regulation and on mammary epithelial cells growth in buffalo. ”

The tables are generally good and the figures are nice and well represent the results of the study.

Authors should check and standardize the references in the list according to journal guidelines.

Authors should check and standardize the references in the list according to journal guidelines.

Author Response

  •  The title well reflects the major findings of the study. However, I suggest to specify the full term of acronym “AGPAT” as “Acylglycerophosphate acyltransferases”

AU: Thank you for comments, we have fixed it according to your comment.

  • Please check the punctuation throughout the text.

AU: Thank you for comments, we have followed your suggestion.

  • The abstract adequately summarize methodology, results, and significance of the study. However, Authors should indicate the statistical analysis applied on the obtained data as well as the results together with P values.   

AU: Thank you for comments, we have fixed them.

  • Please rewrote the sentence “Our data provide new insights into the AGPAT family members on how they regulate the TAG synthesis and growth of mammary epithelial cells in buffalo.”

AU: Thank you for comments. This sentence was modified as “Our data confirmed both AGPAT1 and AGPAT6 could regulate the TAG synthesis and growth of mammary epithelial cells in buffalo”.

  • The introduction section is well written and it falls within the topic of the study. However, more information regarding the peculiarity of lactation phase in livestock should be added. Authors should enhance the first sentence by adding more information regarding physiological adaptation of mammals  during life phases as peripartum and reproduction activity.On this regards, at the beginning of introduction, I suggest to add "During particular life phase such as peripartum females mammals have to face with physiological body demands and adaptation. Particularly, lactation is a very demanding period for the animal, which requires a significant effort of the organism to meet all needs, but also the effort of farmers to provide those needs. Despite the action of homeostatic mechanisms to maintain blood parameters within physiologic levels, changes in metabolites and hormones occur as a result of increased metabolic demands in lactating animals. These changes are not necessarily indicative of diseases but make animals physiologically unstable and more susceptible to a number of metabolic diseases at this stage than during other life periods compromising productivity (Piccione G. et al., Animal Science Paper and Reports, 2009, 27 (4): 321-330; Gianesella M., et al., Journal of Dairy Research, 2019, 86 (3): 291-295; Arfuso F., et al., Theriogenology, 2016, 86: 1156-1164; Fiore E., et al., PLoS One, 2018, 13(4): e0193803; Fiore E., et al. Animal Production Science, 2017, 57 (6): 1007-1013; Morgante M., et al., Animal Science Journal, 2010, 81: 722-730.; Arfuso et al., Archives Animal Breeding, 2016, 59: 429-434)."

AU: Thank you for comments. Although the physiological state of a particular period plays an important role in the impact of animal productivity, our study focused on the expression of related genes in the process of milk fat metabolism. Therefore, we introduced more information about the production and quality of buffalo milk. Because the metabolism of triglycerides is crucial for milk production. For this reason, we have not adopted your suggestion in the revised manuscript. Nonetheless, we appreciate your constructive comments.

  • The section of Materials and Methods is clear for the reader and it meticulously describes the methods applied in the study. However, Authors should check this section and correct many punctuation errors.

AU: Thank you for comments, we have followed your suggestion.

  • Regarding statistical analysis, did the Authors apply a normality test on data in order to assess their normal distribution? Please clarify this aspect.

AU: Thank you for comments. Yes, the normality test was used in the present study.

  • Results section as well as Discussion section is clear and well written. The findings obtained in the study were well discussed and justified with appropriate references.

AU: Thank you for comments.

  • The conclusion section is well written, Authors well summarize the results and the significance of the study. However, I suggest to avoid making references to figures in this section and the use of personal form (i.e. our, we…).

AU: Thank you for comments, we have followed your suggestion.

  • Please delete “In conclusion…” at the beginning of section.

AU: Thank you for comments, we have deleted this phase.

  • I suggest to rewrote the sentence as “According to the findings herein obtained, a total of 13 AGPAT genes were found in buffalo, of which 12 orthologous gene pairs were observed between buffalo and cattle. Furthermore, both AGPAT1 and AGPAT6 were highly expressed in milk samples in buffalo and cattle during lactation. According to in vitro assay applied in the study, both AGPAT1 and AGPAT6 genes are not just regulating the TAG synthesis in mammary epithelial cells but also affecting their growth. These findings herein gathered provided new insight into the role of AGPAT family members on TAG synthesis regulation and on mammary epithelial cells growth in buffalo. ”

AU: Thank you for comments, we have followed your suggestion.

  • The tables are generally good and the figures are nice and well represent the results of the study.

AU: Thank you for comments.

  • Authors should check and standardize the references in the list according to journal guidelines.

AU: Thank you for comments.

  • Authors should check and standardize the references in the list according to journal guidelines.

AU: Thank you for comments.

Reviewer 2 Report

The authors explored different genetic isoforms of The Triacylglycerols (TAG) gene family in Buffalo, with specific focus on their role in milk synthesis. Further comparison conducted against the Bovine TAG isoforms. Three of the identified 13 genes belongs to this family were found to be abundent in Buffalo. The authors further followed on the top two isoforms AGPAT1 and AGPAT6, by performing proliferation experiments of knockdown and control  expression in mammary cell culture from Buffalo and Cattle.

Overall the manuscript is well written and the scientific experiments are sound.

Major points that need attention are:

1. Amongst the sequences from different species, both the river and the swamp buffalo were mentioned - what were the genetic differences between the isoform from these two related sub-species?

2, You decided to follow only on the top two isoforms and not the third relatively over-expressed isoform, LPCAT1 which not seems to be present in cattle. Why? Is it possible that this isoform is perhaps the cause for the different in milk content percentage between cattle and Buffalo? I suggest to add a discussion paragraph on its function.

3. What is the implications of the different isoforms in buffalo to the milk production industry i.e. can you link it to the 77% less cholesterol content compared to cattle? (Perhaps add this point to the discussion section)

Other minor comments:

General -

1. Gene names should be listed in full in the first time they introduced, followed by their abbreviation and Gene bank ID.

2. Similarly, please list any abbreviation in full throughout the article, e.g. EGTR (Line 68), PUFA (Line 78); LPA (line 278) & Some abbreviations are listed in full in the methods section, but the results section abbreviations are coming first in your manuscript, e.g. HMMER (Line 95).

- Please revise throughout the manuscript.

3. There are several places where discussion points are presented in the results section, please review this (e.g. Lines 143-144; from Line 159; from Line 180).

Specific comments:

4. Line 50 - "most TAG synthesis is preformed via the de novo pathway..." - it is not clear if this statement is true in general or specifically in bovine/buffalo species?

5. Reword Lines 101 to 103 - revise the use of 'top'/'larger' and 'smaller' correctly. Might be sufficient to change this to: "...divided into four clusters (Cluster I, II, III, and IV) that contained 4,2,2, and 5 isoforms/genes, respectively."

Figure 2A - Very hard to distinguish between the green and blue color boxes. It will be even harder in Black and white printing - consider use of a different color scheme.

Figure 2D is GPAT4 an alternative name to AGPAT6? I believe this is teh first time it is mentioned. Also, this graph represent expression in mammary gland cell culture (not tissues, and not in plural).

Figure 3, 4 & 5 - please explain the abbreviation of "NC" and "si-AGPAT1/6". Consider listing the full names of the genes represented in these figures and abbreviations of "boMEC"/"buMEC" (Fig 3) to have this figure as a clearer stand alone figure.

2nd paragraph in Discussion - I suggest to change the flow/structure of this paragraph - first highlight your findings, then compare to other reports on bovine species.

Line 271 - insert "the" before "de novo..."

Line 290 - replace "Were" with "are"

Line 326 - from which database the specified project numbers are?

Line 351 - remove the (R) symbol

Line 374 - specify the % of the PBS solution.

Author Response

  • Amongst the sequences from different species, both the river and the swamp buffalo were mentioned - what were the genetic differences between the isoform from these two related sub-species?

AU: Thank you for comments. Yes, the river and swamp are the two subspecies of buffalo. The most obvious genetic difference is that the river buffalo has 25 pairs of chromosomes, while the swamp buffalo has 24 pairs of chromosomes. This information has added to the Introduction section.

  • You decided to follow only on the top two isoforms and not the third relatively over-expressed isoform, LPCAT1 which not seems to be present in cattle. Why? Is it possible that this isoform is perhaps the cause for the different in milk content percentage between cattle and Buffalo? I suggest to add a discussion paragraph on its function.

AU: Thank you for comments. milk production traits, including milk fat percentage and milk fat content, are quantitative traits controlled by multiple genes. Meanwhile, gene expression is also a complex process. Therefore, it is difficult for a gene to reflect the difference in milk production between two species, not to mention that the expression level of this gene is not the highest expressed gene in the two species. Nonetheless, we add some information in the Discussion section. The sentences are "We found that the LPCAT1 gene has a higher expression level in buffalo milk than that in dairy cows, which may explain the difference in milk fat content between the two species. However, this needs further experimental verification. "

  • What is the implications of the different isoforms in buffalo to the milk production industry i.e. can you link it to the 77% less cholesterol content compared to cattle? (Perhaps add this point to the discussion section)

AU: Thank you for comments. As we know, milk fat metabolism is a complex process involving the expression and regulation of many genes, such as GPAM, DGAT1 and LIPIN, etc. In the Introduction section, we have stated that most TAG synthesis is performed via the de novo pathway in most cells. Four rate-limiting steps play a vital role in determining the triglyceride content, which are controlled by the glycerophosphate acyltransferase (GPAT), acylglycerophosphate acyltransferase (AGPAT), lipid phosphate phosphohydrolase (LPIN), and diacylglycerol acyltransferase (DGAT) gene families, respectively. Thus, this correlation can be ignored for the time being.

  • Gene names should be listed in full in the first time they introduced, followed by their abbreviation and Gene bank ID.

AU: Thank you for comments, we have followed your suggestion.

  • Similarly, please list any abbreviation in full throughout the article, e.g. EGTR (Line 68), PUFA (Line 78); LPA (line 278) & Some abbreviations are listed in full in the methods section, but the results section abbreviations are coming first in your manuscript, e.g. HMMER (Line 95).

- Please revise throughout the manuscript.

AU: Thank you for comments, we have followed your suggestion. 

  •  There are several places where discussion points are presented in the results section, please review this (e.g. Lines 143-144; from Line 159; from Line 180).

AU: Thank you for comments, We think these sentences can be seen as some linguistic link, and can also be placed in the results section.

  • Line 50 - "most TAG synthesis is preformed via the de novo pathway..." - it is not clear if this statement is true in general or specifically in bovine/buffalo species?

AU: Thank you for comments. This sentence is corrected as “The de novo pathway is the major pathway for TAG synthesis is performed in most mammalian cells. ”

  • Reword Lines 101 to 103 - revise the use of 'top'/'larger' and 'smaller' correctly. Might be sufficient to change this to: "...divided into four clusters (Cluster I, II, III, and IV) that contained 4,2,2, and 5 isoforms/genes, respectively."

AU: Thank you for comments, we have followed your suggestion. This sentence has added in the revised manuscript

  • Figure 2A - Very hard to distinguish between the green and blue color boxes. It will be even harder in Black and white printing - consider use of a different color scheme.

AU: Thank you for comments, we have followed your suggestion.

  • Figure 2D is GPAT4 an alternative name to AGPAT6? I believe this is the first time it is mentioned. Also, this graph represent expression in mammary gland cell culture (not tissues, and not in plural).

AU: Thank you for comments. Yes, the GPAT4 an alternative name to AGPAT6.

  • Figure 3, 4 & 5 - please explain the abbreviation of "NC" and "si-AGPAT1/6". Consider listing the full names of the genes represented in these figures and abbreviations of "boMEC"/"buMEC" (Fig 3) to have this figure as a clearer stand alone figure.

2nd paragraph in Discussion - I suggest to change the flow/structure of this paragraph - first highlight your findings, then compare to other reports on bovine species.

AU: Thank you for comments. We have provided the detail information for them in the Figure legend. Moreover, we have changed the structure of the 2nd paragraph in Discussion

  • Line 271 - insert "the" before "de novo..."

AU: Thank you for comments, we have fixed it.

  • Line 290 - replace "Were" with "are"

AU: Thank you for comments, we have fixed it.

  • Line 326 - from which database the specified project numbers are?

AU: Thank you for comments. The two project both from the NCBI SRA database. Of course, this information has added in the revised manuscript.

  • Line 351 - remove the (R) symbol

AU: Thank you for comments, we have fixed it.

  • Line 374 - specify the % of the PBS solution.

AU: We have added the solution information in the revised manuscript

Reviewer 3 Report

The manuscript presents the interesting data on characterization and role of the buffalo AGPAT family members in the milk fat synthesis pathway using modern diagnostic tools. The study was well designed.

The minor comments

Introduction

Line 48 – a lack of space before the references

Results

Lines 140-141; 143-144 – the two sentences should be transferred to the Discussion section

Line 186 – the authors should specify ‘a significant level’

Discussion

Line 279, 291 – a lack of space before the references

Line 291 – please remove the space before the point  

Materials and Methods

Line 392 – the authors should specify in which conditions the two statistical tests were used?

Author Response

  • Line 48 – a lack of space before the references

AU: Thank you for comments, we have fixed it.

  • Lines 140-141; 143-144 – the two sentences should be transferred to the Discussion section

AU: Thank you for comments, we think the two sentences were more suitable for the results section. 

  • Line 186 – the authors should specify ‘a significant level’

AU: Thank you for comments, the significant level has added.

  • Line 279, 291 – a lack of space before the references

AU: Thank you for comments, we have fixed them.

  • Line 291 – please remove the space before the point  

AU: Thank you for comments, we have fixed it.

  • Line 392 – the authors should specify in which conditions the two statistical tests were used?

AU: Thank you for comments, we have fixed it.